# Drought and Salinity in Citriculture: Optimal Practices to Alleviate Salinity and Water Stress

**Vasileios Ziogas** [1,*], **Georgia Tanou** [2], **Giasemi Morianou** [1] and **Nektarios Kourgialas** [1,*]

1   Institute of Olive Tree Subtropical Crops and Viticulture, ELGO—DIMITRA, 73134 Chania, Greece; morianou@elgo.iosv.gr

2   Institute of Soil and Water Resources, ELGO—DIMITRA, Thermi, 57001 Thessaloniki, Greece; gtanou@swri.gr

*   Correspondence: ziogas@elgo.iosv.gr (V.Z.); kourgialas@elgo.iosv.gr (N.K.)

**Abstract:** Among the various abiotic stresses, drought is the major factor limiting crop productivity worldwide. Citrus has been recognized as a fruit tree crop group of great importance to the global agricultural sector since there are 140 citrus-producing countries worldwide. The majority of citrus-producing areas are subjected to dry and hot summer weather, limited availability of water resources with parallel low-quality irrigation water due to increased salinity regimes. Citrus trees are generally classified as "salt-intolerant" with high water needs, especially during summer. Water scarcity negatively affects plant growth and impairs cell metabolism, affecting the overall tree growth and the quality of produced fruit. Key factors that overall attempt to sustain and withstand the negative effect of salinity and drought stress are the extensive use of rootstocks in citriculture as well as the appropriate agronomical and irrigation practices applied. This review paper emphasizes and summarizes the crucial role of the above factors in the sustainability of citriculture.

**Keywords:** abiotic stresses; citrus water needs; irrigation practices; sustainability of citriculture



## 1. Introduction

Citrus has been recognized as a group of fruit tree crops of great importance for the global agricultural sector. The majority of the cultivated areas of citrus are located in the subtropical region, in the so-called citrus belt, defined by the 40° north–south latitudes, where the temperature rarely drops below severe freezing temperatures [1]. The majority of the citrus-producing areas are subjected to dry and hot summer weather, limited availability of water resources with parallel low-quality irrigation water due to increased salinity regimes. These factors negatively affect citrus tree productivity and fruit quality. Furthermore, the negative effect of climate change in citrus-producing areas should not be neglected since it augments the detrimental effect of salinity and drought stress [2].

Climate change, combined with the resulting desertification and overexploitation of water resources, due to overpopulation and intensification of agriculture, will be a challenge for the survival, growth, and sufficient yield of agricultural commodities [3]. Especially in citrus crops, water scarcity negatively affects plant growth and impairs cell metabolism, affecting the overall tree growth and produced fruit quality. Drought stress is also affecting the post-harvest handling of citrus fruit since it reduces significantly the reed thickness, rendering the fruit more prone to damage during handling and transportation [4].

Salinity and drought stress demonstrate similar physiological disorders to plants when they occur. Under the effect of salinity and drought stress, interlinked molecular responses are activated in order to provide an acclimation effect to the plant and initiate signaling cascades so as to facilitate the alleviation of the occurred stress syndromes [5]. There are several molecular interactions between salinity and drought stress that are impossible to separate in the field. Additionally, overall plant responses to simultaneous stress factors are complex and can be different in terms of response to each individual stress factor, depending also upon the duration and intensity of each stress syndrome [2].

Citrus are generally classified as "salt-intolerant" crops since irrigation with salinized water immediately arrests tree growth and negatively affects fruit quality, more than in many other crops [6]. Citrus at cellular and organism level can cope with salt and water deficit via the implementation of stress avoidance and stress tolerance mechanisms that block ion accumulation and tissue dehydration or maintain the integrity of cell structures and functionality of crucial biomolecules [7].

A key factor to the overall attempt to sustain and withstand the negative effect of salinity and drought stress is the extensive use of rootstocks in citriculture. Extensive work has been conducted upon citrus trees showing that rootstocks are a key component to the ability of the tree to withstand water scarcity since they modulate the physiological performance of the tree via variations in plant hydraulic conductance, leaf water potential, and stomatal conductivity [8,9].

Several reports indicate that citrus growers worldwide encounter several cultivational problems due to drought and salinity stress, which cause the decline in citrus yield and fruit quality. Thus, it is of paramount importance to provide the necessary means and insight that would guide farmers into the implementation of novel agricultural practices that would increase their income and minimize the negative impact of salt and water stress. Towards this goal, several traditional breeding programs are active and have produced several improved varieties of citrus plants [10]. The deleterious effect of water-related abiotic stresses such as salinity and drought stress can be minimized via the precise and sophisticated agricultural practices that take into account the actual tree needs and the availability of natural resources. The scope of the current review is to harness useful scientific data that could be utilized and guide the successful implementation of agronomic practices that improve citrus tolerance against stress factors.

## 2. Citrus Salinity Stress and Responsive Mechanisms

Citrus are cultivated into areas that are characterized by low levels of precipitation, and in most cases additional irrigation is a necessity. The usage of poor-quality irrigation water combined with the extensive dry and hot climates within the citrus cultivation belt exposes citrus to salinization regimes, a deleterious environmental constraint [2]. When the concentration of $Na^+$ ions in the soil solution exceeds 1500 ppm or 25 mM, which is the critical concentration for the growth of cultivated crops, plants are stressed by salinity. In citriculture, values of electrical conductivity (EC) over 3 dS m$^{-1}$ and sodium adsorption ratio (SAR) over 9 in saturated soil extract are characterized as critical for the survival of the cultivation. In addition, chlorine concentration values above 355 ppm are prohibited for growing citrus [11]. Moreover, citrus growth and fruit yield have been reported to be negativity affected under soil salinity of 2 dS m$^{-1}$, while a decrease of 13% fruit yield has been observed per each 1 dS m$^{-1}$ salinity increase above 1.4 dS m$^{-1}$, which is the threshold value of electrical conductivity for saturated soil extract [12]. Furthermore, the threshold levels of salinity in the rhizosphere of orange trees cv. Valencia were reported at ECs of 2.5 to 3.5 dS m$^{-1}$ [13]. In lemon trees of cv. Verna, the toxic threshold for salinity stress syndromes is varied with the used rootstock, whereas for sour orange, Cleopatra mandarin, and macrophylla, the threshold values for response were 1.53, 2.08, and 1.02 dS m$^{-1}$, respectively [14].

### 2.1. Response to Salinity Stress

When plants are exposed to salinity, several changes, driven by osmotic factors, occur in plant physiology. These changes are sudden and temporarily modify plant water status and gradually cause toxic syndromes, due to the gradient accumulation of ions [15]. According to Garcia-Sanchez and Syvertsen [16], the presence of salt in the soil solution inhibits the growth of leaves and roots of citrus, due to the decrease of the osmotic potential, which limits water availability to the plant. Initially, as an early response to the occurring stress, there is a rapid decrease in the growth rate of leaves and roots, but after the activation

of various physiological and biochemical mechanisms, a very rapid partial recovery of the leaf growth rate or complete recovery sometimes, in the case of the roots, can occur [17].

In contrast to halophytes, citrus plants were supposedly characterized by an inefficient ability to allocate saline ions into intercellular cell structures, causing a weak osmotic adjustment and the manifestation of drought stress [18]. Recent data demonstrate that cell osmotic adjustment is a crucial response towards plant survival under salinity stress conditions, even when toxic ions are not efficiently excluded [19]. This means that plants under salinity stress tend to promote root $Na^+$ and $Cl^-$ uptake and accumulate these ions to plant parts, so as to achieve proper osmotic adjustment, with the deleterious possibility of long-term toxicity [20]. It has been observed that citrus plants subjected to salinity stress demonstrate an osmotic pressure that is less deleterious to the plant compared with that caused by drought stress via the use of polyethylene glycol (PEG). This fact leads to the assumption that citrus plants use salt ions, which are accumulated mainly into the vacuole, for their osmotic adjustment in order to avoid water stress [18].

### 2.2. Ion Toxicity Interplay under Salinity Stress

During salinity stress, $Na^+$ and $Cl^-$ ions tend to accumulate inside the plant cells at toxic levels. It is not an easy task to set the limits of toxicity levels for these ions, since several factors need to be taken into account, such as the type of salts (accompanying ions) and the established rootstock/scion combination [21].

A concentration of 10–30 mM $Na^+$ in the root cell solution has been characterized as toxic because it inhibits enzyme activity, while in the leaves it is up to 100 mM for some cases [22]. Regarding the $Cl^-$ ions, levels in citrus plants leaves required to cause toxicity start from 0.7% dry matter [18].

According to Al-Yassin [13], for most plants, $Na^+$ is the main cause of toxicity where $Na^+$ tends to accumulate, in the woody roots and trunk, while $Cl^-$ accumulates mainly in young shoots and leaves, causing necrotic lesions. Citrus plants grown under mild salinity stress demonstrate a mostly osmotic-driven decrease in fruit yield, without any visual severe toxicity symptom due to the accumulation of $Na^+$ or $Cl^-$ ions. Under intense salinity stress, citrus plants accumulate excessive levels of $Na^+$ and $Cl^-$ ions into the canopy, reaching toxicity levels and severely deregulating the photosynthetic apparatus and tree growth [23].

One of the most crucial factors that needs attention is the ability of ion exclusion from shoot tissues, which is orchestrated by the rootstock. It needs to be clarified that $Cl^-$ is not considered more metabolically toxic than $Na^+$ for citrus plants. Citrus plants, like the majority of woody perennial plants, possess the ability to store $Na^+$ in the woody root-sphere and basal stem parts and exclude it from the leaves via xylem retrieval [24]. Thus, the remaining $Cl^-$ ions that cannot be excluded become the most harmful and toxic element of the saline solution [15]. Therefore, the physiological frame that needs to be established in order to examine citrus resistance to salinity stress is interlinked with the ability of the plant to restrict the transportation of $Cl^-$ ions from the root to the scion, a mechanism that is tightly controlled by the rootstock [25]. A typical example is the fact that the ability of Swingle citrumelo rootstock to maintain lower levels of $Na^+$ in leaves, compared with rough lemon, is due to the ability of the former to sequestrate $Na^+$ in root tissue vacuoles and immobilize them into the cell wall [26].

Scientific results justify the fact that in citrus, $Cl^-$ ions are involved in the deleterious effect of leaf necrosis, growth arrest, and leaf abscission [27]. Leaf abscission is driven by the endogenous levels of phytohormone abscisic acid (ABA) and 1-aminocycloprpane-1-carboxylic acid (ACC), which demonstrated a gradual increase after the establishment of the salinity stress factor [28]. Moreover, other molecules such as polyamines have been proposed as signaling molecules during the adjustment of citrus plants to salinity stress [29].

## 2.3. Salinity Avoidance Mechanisms

Plants employ multiple strategies in order to endure salt stress. These include adjustment mechanism, which is exerted via the accumulation of hormones such as ABA, osmotic adjustment, preferential accumulation of ions into the vacuole, maintenance of photosynthesis via the activation of the water–water cycle [30], the activation of the antioxidant machinery [31], photorespiration, and glycolate oxidase and salt exclusion [17]. These salt stress responses are tightly genotype-specific and influenced greatly by the growth stage of the plant, the rootstock, and the implemented agricultural practices [32].

In citrus plants, salt stress via the accumulation of $Cl^-$ ions trigger the biosynthesis of ethylene precursor (ACC). The prolonged exposure of citrus plants to salinity regimes accelerates ethylene-driven leaf abscission. As an outcome, salinized citrus plants exert ABA accumulation in order to counteract ethylene upsurge. It has been established that citrus plants pretreatment with ABA reduces ethylene release and leaf abscission via the prevention of $Cl^-$ accumulation in leaves [27].

In addition, citrus plants as glycophytes achieve osmotic adjustment via the synthesis of compatible solutes such as proline, sugars, and organic acids. The dominant employed strategy depends upon the genotype (species—cultivar), growth conditions, and implemented agronomic practices. Proline is considered among the most important osmolyte in salt-stressed plants. Experimental data state that the use of exogenous proline (5 mM) significantly minimized the negative impact of sanity stress in salt-sensitive orange cv. Valencia cell lines when exposed to 100 mM sodium chloride (NaCl) [33]. Additionally, sucrose, glucose, and fructose concentrations declined in leaves of Cleopatra mandarin, and in both leaves and roots of Troyer cintrages under continuously increased levels of salinity stress (0–80 mM), suggesting that sugar levels tend to increase in salt-sensitive genotypes rather than in the resistant ones [34].

Furthermore, citrus plants exert their ability to cope with salinity stress via the preferential accumulation of $K^+$ in leaf and stem tissues. It has been documented that regardless of the used rootstock, grafted lemon trees of cv. Fino 49 exhibited lower leaf $Cl^-$ but higher $K^+$ when 10 mM potassium chloride was added to the 50 mM NaCl salinity-induced solution [35]. Moreover, the crucial role of $K^+$, proline, and monosaccharides in osmoregulation under salinity stress syndromes (100 mM NaCl) was suggested due to the significant lower leaf $Na^+$ and $Ca^{2+}$ levels but higher $K^+$, glucose, fructose, and proline concentrations when Trifoliate orange seedlings were inoculated with two mycorrhizal fungi [36].

In citrus plants salinity resistance is also implemented via ion exclusion procedures. Salt exclusion describes the ability of the roots and/or basal stem tissues to translocate low portions of deleterious salts to the photosynthetic active leaves. Genetic ploidy level influences the relative salt resistance of rootstocks, and tetraploid citrus seedlings demonstrate greater salt resistance than that of diploid genotype [37]. In-depth molecular analysis revealed that in citrus plants, the $Na^+$ and $Cl^-$ exclusion mechanism is a heritable characteristic, a fact that led to the establishment of several development programs that focused upon the production of citrus hybrids that exclude salinity ions efficiently and perform even better than the parent genotypes [38].

## 2.4. Tolerance Mechanisms

Salt stress is also linked to the hazardous effect of salt ions absorption, which can harm numerous subcellular organelles, primarily mitochondria and chloroplasts, when they accumulate to high levels in the cells. These ions can also have a deleterious impact on several enzyme processes. In salt stress, tolerance mechanisms are thus techniques for including and excluding harmful ions and protecting crucial metabolic pathways. According to Zhang et al. [39], at the level of the photosynthetic mechanism, salinity decreases the activity of phosphoenolpyruvate carboxylase (PEPC), while the inhibition of the activity of enzymes linked with the Calvin–Benson cycle (ribulose 1,5-diphosphate carboxylase, ribulose 5-phosphate kinase, ribulose 5-phosphate isomerase, and glyceraldehyde 3-phosphate dehydrogenase) has been also documented [40]. Scientific data demonstrate that damage

to the photosynthetic apparatus, under salinity conditions, leads to impairment of the electron transport chain, which leads to an increased rate of free radical production—reactive oxygen species (ROS) and reactive nitrogen species (RNS) [41–43]. It is well established that citrus plants under salinity stress do respond in a positive manner towards the establishment of a sufficient antioxidant defense arsenal in order to cope with the deleterious effect of ROS- and RNS-mediated attacks [44–46].

Plants under salinity stress induce the biosynthesis of specific protein groups, hydrophilins, and heat shock proteins (HSPs), which protect cell compartments and contribute to the overall plant cell protection of vegetative tissues. In vegetative tissues, under optimal growth conditions, these two groups of proteins are in most cases untraceable, but under the effect of salinity stress, ABA-dependent signaling cascades trigger their accumulation [47]. The late embryogenensis abundant proteins (LEA proteins) belong to the functional group of hydrophilins that facilitate cell survival under severe deprivation of water from salinity, even to plants such as citrus [48].

In plants under salt stress, the coordinated action of the $Na^+$ transporters HKT1 and SOS1 regulates $Na^+$ and $K^+$ homeostasis [49]. Shi et al. [50] found that SOS1 enhances $Na^+$ exclusion by extruding the cation from root tip epidermal cells. SOS1 loads $Na^+$ into the xylem sap in xylem parenchyma cells, allowing for effective osmotic adjustment of shoot tissues via vacuolar compartmentalization, whereas HKT1 mediates the reverse flux, unloading $Na^+$ from xylem vessels to prevent $Na^+$ overaccumulation in photosynthetic organs [51]. The coordination of both the HKT1 and SOS1 $Na^+$ transporters allow for proper cation partitioning between organs. HKT1 can transfer sodium recovered from xylem sap into the phloem for shoot-to-root transfer. The excess $Na^+$ recovered by HKT1 from the root and stem xylem sap may eventually represent a significant component of the $Na^+$ trapped in woody tissues in woody perennial plants like citrus. The expression of the SOS1 and HKT1 genes has been linked to the ability of two genotypes, Cleopatra mandarin and trifoliate orange, to exclude $Na^+$ [52].

*2.5. Genetic Approaches to Improve Salinity Stress*

Genetic improvement toward salinity in citrus plants can be implemented either via the identification of natural variations by direct selection or by quantitative trait loci (QTL) mapping. Due to the limited success of direct selection in open field conditions, the interest of breeders has focused upon the identification of genes and gene products that can be transferred, in order to create cultivars via marker-assisted breeding and genetic transformation [32]. QTLs are genomic stretches (section of DNA) that correlate with a variation of a quantitative trait in the phenotype of a population of organisms [53]. In plants, such quantitative variations could be the result of the combined action of multiple different genes and environmental stimuli. QTL analysis involves crossing two parents differing in one or more quantitative traits, so as to identify candidate genes underlying the desired trait [54]. A total of 98 QTLs putatively linked to salinity resistance were determined in a hybrid citrus population (Cleopatra mandarin (salt-tolerant) × Trifoliate Orange (salt-sensitive)). A specific cluster of QTLs controlling plant vigor and leaf boron concentration pointed to a genomic region in linkage group 3 as the most relevant one that could be used in order to improve salinity tolerance using Cleopatra mandarin as a donor [55]. Additionally, QTL mapping revealed 70 potent QTLs in a BC1 population (*Citrus grandis* × (*Citrus grandis* × *Poncirus trifoliata*)), of which 69% were related to salinity. In-depth analysis of 16 regions of the citrus genome revealed that six of them were linked with both plant growth and dry mass production under salinity stress [56].

## 3. Citrus Drought Stress and Responsive Mechanisms

Considering the impact of abiotic stresses upon tree physiology and productivity at a global scale, drought stress is characterized as one of the most deleterious stress factors. Drought impairs normal growth, disturbs water relations, and reduces water use efficiency in plants. Plants, however, have a variety of physiological and biochemical responses at

cellular and whole-organism levels, making it a more complex phenomenon. The rate of photosynthesis is reduced mainly by stomatal closure, membrane damage, and disturbed activity of various enzymes, especially those involved in ATP synthesis [51].

Citrus trees under drought stress demonstrate a significant reduction of growth and cellular metabolic processes, with a concomitant reduction in crop yield and fruit quality [57]. Under the effect of drought stress, citrus reduces physiological parameters such as stomatal conductance ($g_s$) and net assimilation of $CO_2$ ($A_{CO2}$) and leaf transpiration ($E_{leaf}$) [2]. The ability of citrus to cope with the negative impact of water deprivation is also related to the genotype of the plant, resulting in the following order of drought resistance: good resistance—mandarins (*Citrus reticulata* spp.) > rangpur lime > rough lemon > sour orange > *Citrus macrophylla;* medium resistance: lemon > trifoliate orange > citrange hybrid > *Citrus chuana;* poor tolerance: sweet orange > *Citrus verrucose* > grapefruit [9,58].

*3.1. Drought Stress Resistance Mechanisms*

Plants manage to deal with drought stress conditions through physiological, biochemical, anatomical, and morphological modifications. The plant's first response to drought is to minimize stress, which prevents the accumulation of fluids or harmful ions in sensitive leaf tissues. Avoidance mechanisms alone may be adequate to maintain plant performance in the case of mild stress or stress of short duration [59]. The mechanisms that allow the plant to maintain tissue water potential ($\Psi_w$) and water content near to the unstressed level by enhancing water intake or restricting water loss are referred to as whole-plant water dehydration avoidance. Stomatal closure regulation is a fundamental avoidance mechanism that works in the short term. In the long run, leaf rolling flexibility, increasing the root/shoot ratio by creating a deeper and thicker root system, reducing leaf biomass, increasing cuticular resistance, and regulating root water conductivity may be the most important factors for crop plants [60,61]. Cell dehydration avoidance mechanisms are associated with osmotic adjustment and cell wall hardening responses. Osmotic adjustment by the regulation of leaf osmolites appears to be different from many other plant species since heterogeneous responses indicate that osmolytes accumulation can depend on genotype, intensity, and duration of the stress [62]. Osmoregulation is achieved by the biosynthesis of compatible osmolytes such as proline and other betaines, with the exception of glycine betaine [63]. Among the mechanisms of drought avoidance, citrus has the ability to modify the elasticity of the cell wall. By increasing the elasticity of the cell wall, a reduction of the pressure potential is achieved, contributing to the reduction of the water potential, while maintaining the cell's turgor as the cell shrinks around its contents [64]. In citrus, leaf age affects the plant's response to drought, as older leaves have characteristics that allow them to cope with drought more effectively than younger leaves can [64]. When citrus trees are under the effect of drought stress, the control of stomatal opening is the most crucial factor for their survival. Under water stress conditions, reduced stomatal conductivity ($g_s$), decreased respiration rate (E), and reduced net anabolic rate ($A_{CO2}$) are observed [65]. In the presence of intense water stress, stomatal closure is immediate, with complete inhibition of gas exchange within two hours [66], and inhibition of the assimilation rate of $CO_2$ [63]. Under the influence of water stress, the leaf area is significantly reduced, in relation to the root rhizosphere, resulting in an increase in root/shoot ratio, with parallel growth allocation of the roots to deeper soil layers [67].

Stress tolerance mechanisms become critical for plant survival or efficient recovery from stressful situations if the stress gets more severe and the plant is no longer able to maintain enough water (decrease of $\Psi_w$) or ion homeostasis. Cell dehydration tolerance mechanisms are characterized by the accumulation of osmoprotectants, antioxidants, and reactive oxygen species (ROS) scavengers, as well as the biosynthesis of cell-protecting proteins, such as HSPs and hydrophilins. Plants respond by overproducing antioxidant enzymes such as superoxide dismutase, catalase, and peroxidase, as well as metabolites such as ascorbate and glutathione, to reduce ROS toxicity. The ROS response correlates favorably with the degree of resistance of citrus plants [68,69]. It is commonly acknowledged that

antioxidant defenses are essential for abiotic stress tolerance. The accumulation of proteins that play distinct functions in cell defense, such as hydrophilins and HSPs, is another major mechanism of plant cell tolerance caused in vegetative tissues subjected to water stress. Hydrophilins and HSPs are normally undetectable or very tiny in vegetative tissues, but in response to osmotic stress, a strong ABA-dependent transcriptional activation is produced, resulting in a significant accumulation of these proteins [70,71]. Hydrophilins, such as late embryogenesis abundant (LEA) proteins, help cells survive protoplasmic water depletion. Several studies have shown that hydrophilins and HSP protection proteins have a role in citrus plants' responses to water stress [48,72,73].

### 3.2. Intercellular Signaling Cascade and Control of Gene Expression under Drought Stress

Under conditions of drought, an increase in the concentration of abscisic acid (ABA) in roots and leaves is recorded [74]. The accumulation of ABA is due to the increased rate of its biosynthesis or the arrest of its cleavage reactions. Recent studies have revealed the existence of different receptors that recognize the presence of ABA intracellularly and extracellularly [75]. The sesquiterpene plant hormone binds to PYR/PYL/RCAR receptors in leaves to cause rapid ABA-mediated stomatal closure [76]. PP2C phosphatases (ABI1 and ABI2) suppress autophosphorylation of the $Ca^{2+}$-independent and $Ca^{2+}$-dependent kinases SnRK2.6/OST1 and CPKs, respectively, in the absence of ABA, making them inactive. When RCAR/PYR/PYL receptors in guard cells detect ABA, ABI1 and ABI2 phosphatases bind to the ABA–receptor complex and inactivate, releasing SnRK2.6/OST1 and causing CPK inhibition. The kinases then phosphorylate and activate the anion channels SLAC1 and SLAH3 (the same anion channels involved in root-to-shoot xylem translocation), depolarizing the guard cell membrane potential and triggering the cascade of events that lead to guard cell turgor pressure loss and stomatal closure. The role of PP2C as a bona fide coreceptor required to boost ABA-binding affinity has been hypothesized as a result of structural studies performed with *Citrus sinensis* (sweet orange) and *Solanum lycopersicum* (tomato) ABA receptors [77].

The ABA slow response modulates signaling pathways that lead to transcriptional control of many genes, including dehydration tolerance mechanisms. These signaling cascades trigger the synthesis of a series of metabolites involved in the stabilization of enzyme complexes, plasma membrane protection, and regulation of osmotic potential in order to preserve the cell turgor [78]. Plant responses to drought are aimed at reducing cell water loss, protecting intracellular structures and molecules, and repairing damage caused by free radicals [79]. Under drought conditions, the ability of the cell to minimize water loss is achieved by regulating the water potential within the cell structure. In drought conditions, the regulation of water potential is considered a vital component to the survival of the plant, as it allows it to maintain its metabolic activities [80]. Dehydration has a minimal effect on the transcriptional regulation of the CsRCAR/PYR/PYL ABA receptors and the CsSnRK2 activating kinases in citrus leaves, although water shortage induces the genes encoding for the clade-A PP2C phosphatases CsABI1, CsAHG1, and CsAHG3 [81].

In citrus plants under severe drought stress, several plant organs are abscised, including leaves, stems, or even fruits. This detrimental effect is driven by the evoked production of the plant hormone ethylene that participates in the fruit and leaf abscission processes with the parallel accumulation of ABA [82–84]. In addition, recent scientific data pinpoint the novel role of jasmonic acid (JA) as a key regulator of water stress alleviation to citrus plants. In citrus plants under drought stress, scientific data reveal an interplay among JA and ABA accumulation, since JA acts upstream towards ABA biosynthesis, thus facilitating the orchestration of physiological responses [85,86]. Furthermore, recent data highlight the role of a novel gasotransmitter molecule, hydrogen sulfide, acting upstream towards the expression of genes and protein synthesis related to the adaptation of citrus plants to forthcoming drought stress syndromes [45]. The crucial role of several ROS, RNS, and chemical agents as priming agents that can potentially trigger intercellular metabolic reactions to-

wards the adaptation of citrus to drought stress has been well established [87]. Table 1 summarizes the main findings/information provided by the above-presented sections.

**Table 1.** Citriculture and salinity/drought stress highlights.

| Salinity Stress | Reference |
|---|---|
| **Citriculture and salinization limits** | |
| Values of electrical conductivity (EC) over 3 dS m$^{-1}$ and sodium adsorption ratio (SAR) over 9 in saturated soil extract are characterized as critical for the survival of the cultivation | [11] |
| Chlorine concentration values above 355 ppm are prohibited for growing citrus | [11] |
| The Cl$^-$ ion levels in citrus plants leaves required to cause toxicity start from 0.7% dry matter | [18] |
| **Response to Salinity Stress** | |
| Inefficient ability to allocate saline ions into intercellular cell structures | [18] |
| Promote root Na$^+$ and Cl$^-$ uptake and accumulate these ions to plant parts, so as to achieve proper osmotic adjustment | [20] |
| Use salt ions for their osmotic adjustment in order to avoid water deficit | [18] |
| **Ion toxicity interplay** | |
| Under mild salinity stress, a mostly osmotic-driven decrease in fruit yield is demonstrated, without any visual severe toxicity symptom due to the accumulation of Na$^+$ or Cl$^-$ ions | [23] |
| Under intense salinity stress, citrus plants accumulate excessive levels of Na$^+$ and Cl$^-$ ions into the canopy, reaching toxicity levels and severely deregulating the photosynthetic apparatus and tree growth | [23] |
| Leaf abscission is driven by the endogenous levels of phyto-hormone abscisic acid (ABA) and 1-aminocycloprpane-1-carboxylic acid (ACC) | [28] |
| Polyamines have been proposed as signaling molecules during the adjustment of citrus plants towards salinity stress | [29] |
| Store Na$^+$ in the woody root-sphere and basal stem parts and exclude it from the leaves via xylem retrieval | [24] |
| Sequestrate Na+ in root tissue vacuoles and immobilize them into the cell wall | [26] |
| **Amelioration mechanisms** | |
| Pretreatment with ABA reduces ethylene release and leaf abscission via the prevention of Cl$^-$ accumulation in leaves | [27] |
| Use of exogenous proline (5 mM) significantly minimized the negative impact of sanity stress in salt-sensitive orange cv. Valencia Late | [33] |
| Genetic ploid level influences the relative salt tolerance of rootstocks, and tetraploid citrus seedlings demonstrate greater salt resistance than shown by diploid genotypes | [37] |
| The Na$^+$ and Cl$^-$ exclusion mechanism is a heritable characteristic—production of citrus hybrids that exclude salinity ions efficiently and perform even better than the parent genotypes | [38] |
| **Enzyme activity and protein synthesis** | |
| Citrus plants under salinity stress do respond in a positive manner towards the establishment of a sufficient antioxidant defense arsenal | [44–46] |
| Induce the biosynthesis of specific protein groups, hydrophilins and heat shock proteins (HSPs), that protect cell compartments and contribute to the overall plant cell protection | [48] |
| **Genetic approaches** | |
| Identification of natural variations by direct selection or by quantitative trait loci—QTL mapping | [55] |
| **Drought Stress** | |
| Reduces physiological parameters such as stomatal conductance ($g_s$) and net assimilation of $CO_2$ ($A_{CO2}$) and leaf transpiration ($E_{leaf}$) | [2] |
| Drought resistance: good resistance—mandarins (*Citrus reticulata* spp.) > rangpur lime > rough lemon > sour orange > *Citrus macrophylla*; medium resistance: lemon > trifoliate orange > citrange hybrid > *Citrus chuana*; poor resistance: Sweet orange > *Citrus verrucose* > grapefruit | [9,58] |
| **Mitigation mechanisms** | |
| Osmolyte accumulation depends upon genotype, and intensity and duration of the stress | [62] |
| Citrus have the ability to modify the elasticity of the cell wall | [64] |
| Leaf area is reduced, with parallel increase of root/shoot ratio and allocation of roots to deeper soil layers | [67] |
| Biosynthesis of compatible osmolytes such as proline and other betaines, with the exception of glycine betaine | [63] |
| Accumulation of osmoprotectants, antioxidants, ROS scavengers, and cell-protective proteins, such as HSP and hydrophilins | [68,69] |
| **Intercellular signaling cascade and control of gene expression** | |
| ABA modulates signaling pathways that lead to transcriptional control of many genes that control the synthesis of metabolites involved in the stabilization of enzyme complexes, plasma membrane protection, and osmotic potential regulation | [78] |

**Table 1.** *Cont.*

| Salinity Stress | Reference |
| --- | --- |
| Under severe drought stress, several plant organs are abscised, including leaves, stems, or even fruits, due to the production of ethylene that participates in the fruit and leaf abscission processes with the parallel accumulation of ABA | [82–84] |
| There is an active interplay between JA and ABA accumulation, since JA act upstream towards ABA biosynthesis, thus facilitating the orchestration of physiological responses | [85] |
| Hydrogen sulfide, acting upstream towards the expression of genes and protein synthesis related with the adaptation of citrus plants | [45] |
| ROS, RNS, and chemical agents as priming agents that can potentially trigger intercellular metabolic reactions towards the adaptation of citrus to drought stress | [88] |

## 4. Agricultural and Irrigation Practices That Cope with Salinity and Drought in Citrus

Figure 1 depicts specific agricultural practices in citriculture that could significantly alleviate the negative effects either from salinity or from drought stress. All these practices are analyzed in the following parts of this section. Citrus trees' need for water depends on tree age, tree size, citrus species, climate, and soil type. As a general guide, research studies suggest that mature citrus (orange) trees need about 4000–5000 m$^3$ of water per hectare and year [89]. Water use for grapefruit and lemon trees is about 20% higher than that of oranges, while water use for mandarins is about 10% less. Additionally, regarding the appropriate irrigation frequency, this depends on the season and soil type and ranges from 7 to 25 interval days [4,87]. There are three main irrigation systems used in citriculture: micro-sprinkler and sprinkler irrigation, which give good results mainly on sandy soils, and drip irrigation, one of the best techniques from a technical point of view, due to water economy. The irrigation schedule should start according to the soil moisture, which can be determined by soil samples with an auger or soil moisture sensors/tensiometers. The installation of tensiometers following the irrigation line in the middle point between two emitters (0.20 m from the emitter, 1 m from the trunk) is recommended for mature citrus trees. In the case of a micro-sprinkler, the installation of tensiometers at a distance of 0.5 m from the mini sprinkler or 1 m from the sprinkler is recommended [90]. Citrus has a relatively shallow root system. Thus, it is important to apply irrigation at the effective root zone, minimizing the deep percolation of water. For citrus, the effective root zone is usually up to 30 cm soil depth (depending on the soil type). At that depth, there is a minimum threshold of critical matric potential, which needs to be maintained. Typical thresholds are 20 kPa for sandy soils and 100 kPa for clayey soils [89,91].

Citrus is moderately resistant of salty water, but salts can accumulate in the soil or on the foliage and can cause root dieback or leaf loss. Salinity will always be more of a problem on poorly drained clay or silt soils than on permeable sandy or gravelly soils. In the case that citrus trees are irrigated by drip or micro-sprinkler method, which is the most common one, care should be taken so that water will not contact the leaves, since salt may burn citrus foliage. Salts may also plug emitter orifices. Frequent or shallow irrigations will lead to salt accumulation on the soil surface (in the form of a white crust), and accumulation in the root zone. Salt may be moved from the root zone through the process of leaching. To leach salty soil, apply large amounts of water to the soil once or twice a year [11].

In citrus, the choice of the rootstock plays a crucial role in the water relations of the tree with the soil and determines its overall response to abiotic stresses [38,64,92]. The genetic characteristics of the rootstock determine the robustness of the scion and its survival under water stress [53]. Studies have shown that orange trees, of cv. Lane Late, grafted on Cleopatra's mandarin rootstock demonstrated in their foliage higher water use efficiency (WUE) and osmotic adjustment and kept this parameter at high levels under drought conditions, compared to plants crafted upon Carrizo citrange hybrid rootstock [58,63].

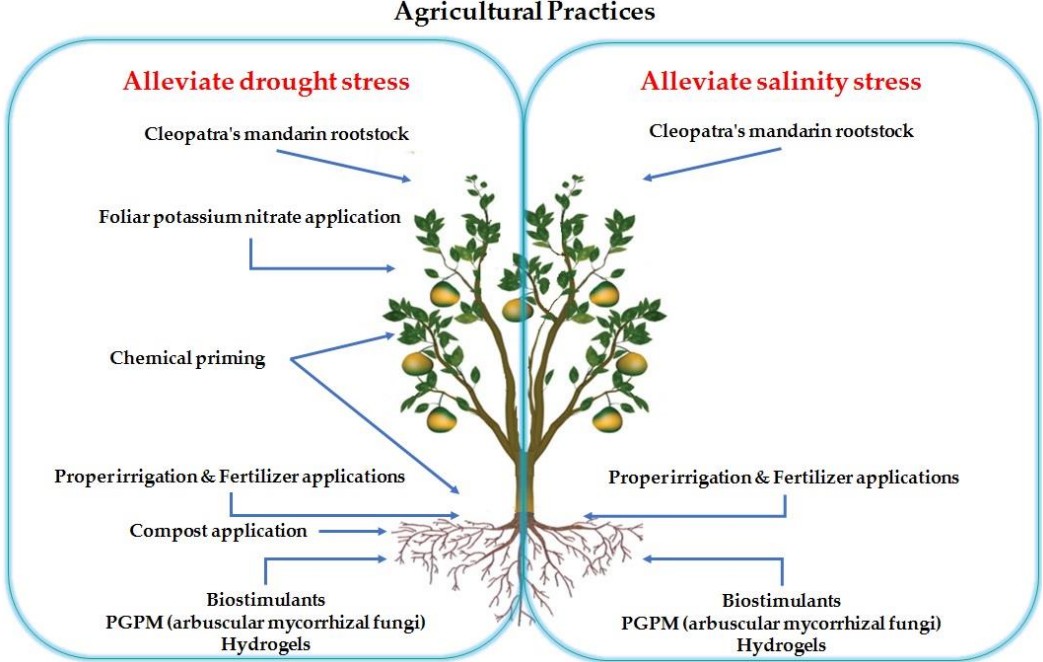

**Figure 1.** A mechanism diagram of agricultural measures to alleviate citrus drought and salinity.

Different relieving methods have been assessed to increase yield under salt stress and water deficit conditions. First, wise control of irrigation and fertilizer applications is considered essential to tackle this problem in a commercial orchard. Other horticultural practices such as the use of orange varieties as interstocks, the use of hydrogels, shading, treatments with persistent analogs of phytohormone abscisic acid (ABA), and polyamines [93] improved citrus performance under drought and/or salt stress conditions, as well. Various practices have dealt with the effect of increasing calcium or nitrates in the nutritional solution as well as treatments in order to relieve salt stress. In the presence of an adequate concentration of calcium ($Ca^{2+}$), plants exclude $Na^+$ more efficiently and avoid their accumulation in cells [94]. It has been repeatedly shown that nitrate and other nitrogen-derived compounds such as urea or ammonium have had positive effects on the growth response of citrus [19]. Moreover, nitrate seems to have two separate effects that can improve the performance of citrus seedlings under saline conditions. First, it has been observed that nitrate supplementation stimulated photosynthesis and growth as well as reduced leaf abscission. Second, the nitrogen-induced increase in leaf biomass has resulted in chloride dilution, the critical factor for salt damage [95]. In addition, foliar potassium nitrate application can improve the endurance of citrus seedlings to drought conditions [93].

Using compost is one of the best ways to conserve irrigation water by retaining soil moisture within the root zone. Apply two to four inches of compost under the plant canopy. Compost can consist of pine needles, leaves, bark, wood chips, straw, or any other organic materials. The compost should not directly contact the trunk and should be expanded as the plant grows. A good cover of compost will help to control weeds under the tree canopy, as well as reduce water evaporation. Trees that are enhanced with compost can be irrigated less frequently than those that are not. Composting can also lower soil temperature, thus allowing for better root growth, and will eventually decompose and add significant organic matter to the soil [96,97].

Moreover, chemical priming has been suggested as a promising method in the field of plant stress physiology and crop stress management. Plants are apparently capable of causing stress "memory" or "stress imprinting" following a first stress exposure, which leads to adjustment to later biotic or abiotic stress. Through priming (also known as hardening), plants are able to trigger responses to a range of stresses, providing low-cost

protection in relatively high-stress/pressure conditions [98]. Reactive oxygen species (ROS) in the form of hydrogen peroxide ($H_2O_2$) and reactive nitrogen species (RNS) in the form of nitric oxide ($NO^\bullet$) induce priming toward salinity and drought in citrus plants [89]. These chemical agents, also including sodium nitroprusside, sodium hydrosulfide, melatonin, and polyamines, can potentially result in enhanced resistance in the field against multiple abiotic stresses [29,99,100].

Furthermore, the use of biostimulants has become a common agricultural practice by many farmers. The use of these compounds provides protective effects against abiotic stress factors and contributes positively to overall plant growth [101]. These alleviating effects are exerted via the orchestrated activity of plant hormones, proline, sugars, amino acids, etc. whose production is stimulated by the applied biostimulant [102]. The recent scientific data highlight that the usage of biostimulants facilitates citrus root development or regulates the osmoregulatory mechanisms in plant cells [103]. Orange trees (*Citrus sinensis* L.), when spayed with commercial extract of *Ascophyllum nodosum,* exerted improved water relations and better water use efficiency (WUE) when irrigated with 50% restitution of evapotranspired water [104]. The use of biostimulants is considered an agricultural practice that could contribute positively to the alleviation of drought stress and increase WUE in citrus crops, especially in drought-prone regions where citrus trees are agronomically important but water resources are limited due to urban use and climate change [101].

A novel and promising agricultural practice is the use of compounds enriched with plant-growth-promoting microbes (PGPM). The extended use of chemical fertilizers and pesticides has caused a severe decline in soil quality, hence there is an urgent need to establish and implement agricultural techniques that will sustain agricultural production. Towards this goal, several novel products have been released that engulf the technology of the application of plant-growth-promoting microbes (PGPM) along with mycorrhizal fungi in the root system of the plant, enhancing plant growth and plant protection against abiotic stress conditions [105]. Specifically, it has been reported that Arbuscular Mycorrhizal induced water deficit tolerance of roots of Trifoliate orange by regulating polyamine homeostasis [106], whereas in another study, drought stress conditions stimulated $H^+$-ATPase activity and PtAHA2 gene expression, resulting in nutrient uptake, increased root growth, and lower soil pH microenvironment [107].

Apart from the above, the hydrogel polymer compound seems to be particularly effective to be used as a soil conditioner in citriculture, increasing crop tolerance and growth in drought conditions. Abobatta and Khalifa [108] highlight that medium or high dose of hydrogel composite (1000 to 1500 g/tree) enhances total yield, fruit weight, and fruit quality (fruit content of total soluble solids and total sugars). This may be due to the crucial role of applied longtime hydrogels in increasing water availability and nutrients for citrus trees. Additionally, research studies indicate that hydrogels may minimize the adverse effects of salinity by reducing the levels of salt ions in citrus tissues [109,110]. Specifically, hydrogel composite releases water and nutrient to the trees when the soil surrounding the root zone starts to dry up. Hydrogel materials cause a reduction in irrigation amount as well as intervals by 50%. In addition, it has been proved that hydrogels can increase soil's water-holding capacity up to four times, ensuring safe soil moisture levels as well as nutrients under drought conditions. There are three forms of hydrogel composites containing natural polymers (polysaccharide derivatives), semisynthetic polymers (cellulosic primitive derivatives), and synthetic polymers. Synthetic polymers indicate higher stability under different environmental conditions than shown by natural ones [111]. Table 2 highlights in a strict matter all the above-mentioned practices and their role to alleviate salinity and water stress in citrus

**Table 2.** Summarizing the main agricultural practices to alleviate drought/salinity stress.

| Agricultural Practice | Alleviate Drought Stress | Alleviate Salinity Stress | Reference |
|---|---|---|---|
| Proper irrigation (amount and frequency) based on tree age, tree species, climate, soil type, and/or saline irrigation | P | P | [4,87] |
| Appropriate fertilizer applications (calcium or nitrates in the nutritional solution) | P | P | [94] |
| Foliar potassium nitrate application | P | NP | [93] |
| Use of Cleopatra's mandarin rootstock and hybrids—orange varieties as interstocks | P | P | [38,63,92] |
| Compost application | P | NP | [96,97] |
| Chemical priming (use of sodium nitroprusside, sodium hydrosulfide, melatonin, and polyamines) | P | NP | [29,99,100] |
| The use of biostimulants | P | P | [103,104] |
| Use of plant-growth-promoting microbes (arbuscular mycorrhizal fungi) | P | P | [107,109] |
| The use of hydrogels | P | P | [111] |

P: positive effect; NP: no positive effect.

## 5. Conclusions

In citriculture, water scarcity and salinity negatively affect plant growth and impair cell metabolism, affecting the overall tree growth and the quality of produced fruit. Relevant factors related to plant response to water deficit and salinity stress have been identified in citrus plants. Under salinity stress, salt ions contribute significantly to osmotic adjustment, so that leaf ion toxicity becomes the main problem. Moreover, resistance to salt stress is mainly associated with the rootstock's ability to exclude chloride. Physiological and molecular approaches, recently supported with omics technologies, have identified different stress resistance mechanisms in various citrus genotypes. A characteristic example of adequate tolerance to salt and drought stress is Cleopatra mandarin citrus rootstocks. High priority should be given to rootstocks that effectively combine stress avoidance and tolerance mechanisms to optimize both plant production under adverse environmental conditions and efficient crop recovery after stress. The recent advance of next-generation sequencing tools and the implementation of omics-oriented technologies could contribute towards the establishment of functional genomics and the in-depth exploration of the gene code of the agricultural important citrus cultivars. Future work should focus upon the in-depth exploration of the impact of optimal agricultural practices towards the alleviation of abiotic stress factors via omic approaches, so as to understand the alleviation strategy that is encoded in each citrus species. The fact that the application of a stress-tolerant microbial consortium of PGPM strains, mycorrhizal fungi, biostimulant compounds, and hydrogels enhances plant growth under abiotic stress conditions pinpoints them as key factors that can solve future food security problems and also maintain soil fertility and plant health.

**Author Contributions:** Writing—original draft preparation, V.Z. and N.K.; review and editing, G.T. and G.M.; supervision, V.Z. and N.K. All authors have read and agreed to the published version of the manuscript.

**Funding:** This research received no external funding.

**Institutional Review Board Statement:** Not applicable.

**Informed Consent Statement:** Not applicable.

**Data Availability Statement:** Data sharing not applicable.

**Conflicts of Interest:** The authors declare no conflict of interest.

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
