# Peer review of "Drought and Salinity in Citriculture: Optimal Practices to Alleviate Salinity and Water Stress"

_agronomy, doi:10.3390/agronomy11071283_

Round 1

Reviewer 1 Report

I my opinion the paper is not well written and needs improvement and many corrections.

Authors incorrectly apply such terms as: adaptation, stress tolerance, stress resistance.

The terms stress resistance and stress tolerance should not be used interchangeably. It is not the same. It should be distinguished. Plants are resistant to stress thanks to evolutionary formed adaptive traits and thanks to ability to adjust to stress conditions (acclimation) by triggering stress avoidance and stress resistance mechanisms.

Moreover, part of the work that describes the effect of both stresses on plants and the mechanisms of resistance should be rewritten. This part is not clearly written. Negative effects of stress should be separated from resistance mechanisms. At the beginning it should be described the negative impact of these stress factors on plants (at metabolic, physiological and all plant level) considering citrus plants. Stress avoidance and stress tolerance mechanisms should be than described including and underlining those found in citrus plants. The involvement of plant hormones in resistance mechanisms should also be correctly described.

In many parts of manuscript some information are repeated without cause.

The last part, that is the conclusions are rather summary and after changes  and shortening should be an abstract.

Others comments and correction were marked in attached manuscript.

Author Response

Response to Reviewer #1:

Comment 1: I my opinion the paper is not well written and needs improvement and many corrections.

Authors incorrectly apply such terms as: adaptation, stress tolerance, stress resistance. The terms stress resistance and stress tolerance should not be used interchangeably. It is not the same. It should be distinguished. Plants are resistant to stress thanks to evolutionary formed adaptive traits and thanks to ability to adjust to stress conditions (acclimation) by triggering stress avoidance and stress resistance mechanisms.

Reply: The terms adaptation, stress tolerance, stress resistance have been revised throughout the text. The term stress resistance has been eliminated.

Comment 2: Moreover, part of the work that describes the effect of both stresses on plants and the mechanisms of resistance should be rewritten. This part is not clearly written. Negative effects of stress should be separated from resistance mechanisms. At the beginning it should be described the negative impact of these stress factors on plants (at metabolic, physiological and all plant level) considering citrus plants. Stress avoidance and stress tolerance mechanisms should be than described including and underlining those found in citrus plants. The involvement of plant hormones in resistance mechanisms should also be correctly described.

In many parts of manuscript some information are repeated without cause.

Reply: Thank you for your comments which was taken into consideration, also repeated information have been erased.

Reviewer 2 Report

In this paper, the damage of drought and salinity to citrus and the mechanism of resistance were reviewed, and some mitigation agricultural measures were also reviewed. Strictly , there are many reviews in this field. However, the foothold of this paper is agricultural measures, so it is necessary. However, I think there are some problems in the article before accepted for publish.    

  (1) The references are too old. As I mentioned earlier, there have been a lot of reviews on the physiological responses of citrus to salinity and drought. However, due to the old references in this paper, some new references have not been summarized and applied. New references must be added.      

  (2) The elaboration of some agricultural measures is not complete. For example, in this article, the authors involved PGPM alleviating citrus stress. As far as I know, in the web of science, there are many related articles. The 94 and 95 references used by the author are published in 2008 and 2014, which are relatively old. In the web of science data, I find two recent papers (Zou YN, et al. 2021, doi: 10.3389/fpls.2020.600792; Cheng et al. 2021, doi:10.3389/fpls.2021.659594) for your consideration. 
 (3) In fact, the authors involve many agricultural and chemical measures to alleviate citrus drought and salinity. I strongly recommend the authors to build a mechanism diagram (please add a figure) so that readers can understand these benefits more quickly.  

  (4) The physiological response of citrus to salinity and drought stress was more than that of molecular mechanism in this review. It is suggested to increase part molecular references to develop the article.     

    (5) Some of the first level titles in the article, such as 2 Salinity stress, can be expanded appropriately to better summarize the content of the chapter. 

Author Response

Comment 1: In this paper, the damage of drought and salinity to citrus and the mechanism of resistance were reviewed, and some mitigation agricultural measures were also reviewed. Strictly, there are many reviews in this field. However, the foothold of this paper is agricultural measures, so it is necessary. However, I think there are some problems in the article before accepted for publish.    

Reply: We thank reviewer for his/her comment.

Comment 2: The references are too old. As I mentioned earlier, there have been a lot of reviews on the physiological responses of citrus to salinity and drought. However, due to the old references in this paper, some new references have not been summarized and applied. New references must be added.    

Reply: Old references have been replaced by recent ones (Haman et al. 2021; Zhang et al., 2021; Acosta-Motos et al., 2017; Li et al., 2020)

Comment 3: The elaboration of some agricultural measures is not complete. For example, in this article, the authors involved PGPM alleviating citrus stress. As far as I know, in the web of science, there are many related articles. The 94 and 95 references used by the author are published in 2008 and 2014, which are relatively old. In the web of science data, I find two recent papers (Zou YN, et al. 2021, doi: 10.3389/fpls.2020.600792; Cheng et al. 2021, doi:10.3389/fpls.2021.659594) for your consideration. 

Reply:  The suggestion was taken into consideration, the context of the proposed agricultural measures was checked and completed, accordingly. The references 94 and 95 have been replaced by the proposed studies as well as the in-text reviewing.

Comment 4: In fact, the authors involve many agricultural and chemical measures to alleviate citrus drought and salinity. I strongly recommend the authors to build a mechanism diagram (please add a figure) so that readers can understand these benefits more quickly.  

Reply:   A mechanism diagram of agricultural measures to alleviate citrus drought and salinity is presented at Figure 1.

Comment 5: The physiological response of citrus to salinity and drought stress was more than that of molecular mechanism in this review. It is suggested to increase part molecular references to develop the article.     

Reply:  Thank you for your comment, new information and references were added in the text.

Comment 6: Some of the first level titles in the article, such as 2 Salinity stress, can be expanded appropriately to better summarize the content of the chapter. 

Reply:  Thank you for pointing this out.  ‘2’ and ‘3’ first level titles have been revised according to reviewer comment.

Reviewer 3 Report

I found the manuscript of Ziogas and colleagues very interesting. The problem of drought afflicts many areas of the planet and over the years, these areas are increasing more and more. Furthermore, the greatest effects are observed in crops of plants for human use. The increase in soil salinity also affects the growth and development of plants and this may be associated with the poor quality of the water used for irrigation. The problem is very complex and the authors address it in many aspects by focusing on Citrus.

The authors discuss many practices used to combat drought and promote the growth of Citrus. They extensively discuss the molecular mechanisms regulated by hormone signaling. Finally, they identify the best agricultural practices to counter these abiotic stresses in Citrus.

Given my great interest in this Review, I suggest to the authors some references for their manuscript:

Naservafaei, S. et al. Biological Response of Lallemantia iberica to Brassinolide Treatment under Different Watering Conditions. Plants 2021, 10, 496. doi: 10.3390/plants10030496

Reza Yousefi, A. et al. Germination and Seedling Growth Responses of Zygophyllum fabago, Salsola kali L. and Atriplex canescens to PEG-Induced Drought Stress. Environments 2020, 7, 107. doi: 10.3390/environments7120107

Mahdavi, A et al. Variation in Terpene Profiles of Thymus vulgaris in Water Deficit Stress Response. Molecules 2020, 25, 1091. doi: 10.3390/molecules25051091

Author Response

Comment 1: I found the manuscript of Ziogas and colleagues very interesting. The problem of drought afflicts many areas of the planet and over the years, these areas are increasing more and more. Furthermore, the greatest effects are observed in crops of plants for human use. The increase in soil salinity also affects the growth and development of plants and this may be associated with the poor quality of the water used for irrigation. The problem is very complex and the authors address it in many aspects by focusing on Citrus. The authors discuss many practices used to combat drought and promote the growth of Citrus. They extensively discuss the molecular mechanisms regulated by hormone signaling. Finally, they identify the best agricultural practices to counter these abiotic stresses in Citrus.

Given my great interest in this Review, I suggest to the authors some references for their

manuscript:

Naservafaei, S. et al. Biological Response of Lallemantia iberica to Brassinolide Treatment under Different Watering Conditions. Plants 2021, 10, 496. doi: 10.3390/plants10030496

Reza Yousefi, A. et al. Germination and Seedling Growth Responses of Zygophyllum fabago, Salsola kali L. and Atriplex canescens to PEG-Induced Drought Stress. Environments 2020, 7, 107. doi: 10.3390/environments7120107

Mahdavi, A et al. Variation in Terpene Profiles of Thymus vulgaris in Water Deficit Stress Response. Molecules 2020, 25, 1091. doi: 10.3390/molecules25051091

Reply:  The author would like to thank the reviewer for his constructive comment. The proposed references were added in the manuscript.

Round 2

Reviewer 1 Report

The manuscript has been improved. However, it still requires corrections and complementations. All comments and suggestion are chacked in attached manuscript.

Author Response

June 18, 2021

Dear Editorial Board of Agronomy/ Reviewer 1,

I am submitting a revised manuscript for consideration of publication in Agronomy, Special Issue “Principle and Practices in Fruit Tree Production and Postharvest Management”. The manuscript is entitled “Drought and salinity in citriculture: Optimal practices to alleviate salinity and water stress”.

The authors tried to consider all your valuable comments which have improved significantly the manuscript.

Responses are summarized in this document and are also highlighted in the manuscript in yellow font.

Response to Reviewer #1:

Comment 1: The manuscript has been improved. However, it still requires corrections and complementation’s. All comments and suggestion are checked in attached manuscript.

Reply: All the prosed changes have been made according to the attached pdf manuscript from reviewer 1. The number of the lines is according to this pdf file provided by reviewer 1.

Line 50. The phrase “In sensitive Plant tissues” has been changed to “and tissue dehydration”

Line 83. The phrase “Tolerance limit to salinity stress” has been changed to “threshold level of salinity”

Line 94. Word “increase” has been changed to “decrease”

Line 95-96.  The lines have been rephrased in order to correspond to the correct scientific information.

Lines 108-109. The proposed phrase “which are accumulated mainly into the vacuole” has been added.

Line 134. Word “tolerance” has been changed to “resistance”

Line 145. Phrase “Adaptive response of citrus plants towards salinity stress” has been changed to “adjustment of citrus plants to salinity stress”

Line 147. Word “amelioration” has been changed with “avoidance”

Line 148. Word “Citrus plants” has been changed with “Plants”

Line 149. Word “adaptation” has been changed with “adjustment”

Line 150-153. The lines have been rephrased to “accumulation of ions into the vacuole, maintenance of photosynthesis via the activation of the water-water cycle, the activation of the antioxidant machinery, photorespiration and glycolate oxidase and salt exclusion.” The paragraph has been corrected and supplemented.

Lines 154-155. The phrase is deleted.

Line 168. word “Late” is deleted

Lines 172, 181, 184, 185. Word tolerant has been changed with “resistance”

Line 190. Title 2.4 has changed to “Tolerance mechanisms”

Line 191. The reference has been corrected, from “Zhu” to “Zhang et al.”

Line 195. “According to various reports” has been changed with “Scientific data demonstrate that”

Line 209. Word “or drought” is deleted

Line 209. Tolerance mechanism to salinity induced dehydration have been added

Line 221. Word tolerant has been changed with “resistance”

Lines 238-240 are deleted.

Line 247, 248, 249, 250. Word “tolerant” has been changed with “resistance”

Line 251. Word “mitigation” has been changed with “resistance”

Line 251. Paragraphs 3.1 and 3.2 have been revised, according to reviewer comments.

Line 252. Word “tolerate” has been changed with “deal with”

Line 257. Word “characterized” has been changed with “achieved”

Line 262. Word “in citrus” is deleted

Line 269. The phrase “reducing the cell's water potential, in order to avoid disruption of cellular” has been changed with “and”.

Line 280. Phrase “abiotic stress such as” is deleted.

Line 311. Table 1. The words “stress” has been changed with “deficit”, the word “adaptation responses” has been changed with “adjustment” and the word “tolerance” has been changed with “resistance”

Line 311. Table 1. Mitigation mechanisms and Intercellular signaling cascade and control gene expression, has been changed according to the text.

Line 313. The phrase “alleviate salinity and water stress” has been changed with “cope with salinity and drought”

Line 341. Word “tolerance” has been changed with “resistance”

Line 356. Word “adaptation” has been changed with “adjustment”

Line 357. The phrase “and tolerate the effects of relatively high NaCl concentration” has been changed with “and avoid their accumulation in cells”

Line 388. Word “adaptation” has been changed with “adjustment”

Line 394, 454, 455, 476. Word “tolerance” has been changed with “resistance”

Line 441. Conclusion section has been shortened, and redundant information has been deleted.

We would like to thank reviewer 1, for the in-depth check of the manuscript. We confirm that this manuscript has not been published elsewhere and that it has not been submitted simultaneously for publication elsewhere. All authors have approved the manuscript and agreed with the submission to Agronomy. The authors have no conflicts of interest to declare.

Thank you very much for your consideration.

Yours Sincerely,

Dr Vasileios Ziogas

Assistant Researcher, Citrus Pomology

Institute of Olive tree, Subtropical Plants and Viticulture

ELGO-DIMITRA, Chania, Crete, Greece

e-mail: ziogas@elgo.iosv.gr